# Numerical Simulation and Observational Data Analysis of Mesoscale Eddy Effects on Surface Waves in the South China Sea

Jin Wang [1,2], Brandon J. Bethel [1], Changming Dong [1,3,4,*], Chunhui Li [1] and Yuhan Cao [1,5]

1   School of Marine Sciences, Nanjing University of Information Science and Technology, Nanjing 210044, China; wangjin002457@nuist.edu.cn (J.W.); 20195109101@nuist.edu.cn (B.J.B.); lichunhui@nuist.edu.cn (C.L.); caoyh@nuist.edu.cn (Y.C.)
2   Nanjing Xinda Institute of Safety and Emergency Management, Nanjing 210044, China
3   Southern Ocean Science and Engineering Guangdong Laboratory (Zhuhai), Zhuhai 519000, China
4   Department of Atmospheric and Oceanic Sciences, University of California, Los Angeles, CA 90095, USA
5   School of Marine Technology and Geomatics, Jiangsu Ocean University, Lianyungang 222005, China
*   Correspondence: cmdong@nuist.edu.cn

**Abstract:** Surface current velocities of mesoscale eddies have a unique annular structure, which can inevitably influence surface wave properties and energy distribution. Sensitivity experiments of ideal mesoscale eddies on waves were carried out by the Simulating WAves Nearshore (SWAN) wave model to investigate these influences. In addition, China–France Oceanography SATellite Surface Wave Investigation and Monitoring (CFOSAT-SWIM) observational data of a large warm-cored eddy in the South China Sea (SCS) during the period of October–November 2019 were used to validate the influence of mesoscale eddies on waves. The results illustrated that mesoscale eddies can alter wave properties (wave height, period, and steepness) by 20–30%. Moreover, wave direction could also be modified by 30°–40°. The current effect on waves (CEW) was more noticeable with strong currents and weak winds, and was governed by wave age and the ratio of wave group velocity to current velocity. Wave spectra clearly indicated that current-induced variability in wave energy distribution happens on a spatial scale of 5–90 km (i.e., the sub- and mesoscales). Through comparing the difference of wave energy on both sides of an eddy perpendicular to the wave propagation direction in an eddy, a simple way to trace the footprints of waves on eddies was devised.

**Keywords:** ocean mesoscale eddies; surface waves; current effects on waves; CFOSAT-SWIM; SWAN

## 1. Introduction

A wide array of both past and recent studies have highlighted that a diverse body of submesoscale and mesoscale phenomena, such as eddies, fronts, and vortex filaments, amongst others, are very active in and ubiquitous throughout the global ocean [1–3]. Phillips [4] and Mei [5] recognized that these phenomena can induce strong variability in surface waves through wave–current interactions. Specifically, when waves interact with non-uniform currents, waves undergo changes in wave steepness, surface, roughness, and wave breaking in addition to the occurrence of extreme wave heights [6,7]. Crucial for this paper, mesoscale eddies have surface current velocities with a unique annular structure which can affect both wave properties and their energy distributions. Through the usage of field measurements and numerical models, the current effect on waves (CEW) has been rigorously explored.

Analyses of Synthetic Aperture Radar (SAR) imagery, satellite altimeter measurements, and airborne observations can be used to document the widespread effect of eddies on surface waves. For example, Antony et al. [8] used SAR images to monitor the imprinting of a mesoscale eddy on a surface wave field in the Gulf of Alaska. Romero et al. [3] used airborne observations to study and catalogue the effects of the Gulf Stream and a nearby

eddy on local waves. It was identified that the nonlinear characteristics of the waves changed significantly after passing through the strong current, and wave breaking rates increased. Additionally, where waves and currents propagated/flowed opposite to one another, the significant wave height ($H_s$) could be modulated by 30%. Quilfen et al. [9] used SAR and satellite altimeters to analyze the wave field across the Agulhas Current and found that although patterns identified in a wave model forced with altimeter-observed surface currents were consistent with SAR measurements, the CEWs were under-predicted.

Numerical simulations into CEW have also been extensively conducted. Recently, Ardhuin et al. [2] used satellite altimetry alongside numerical models to show that the modulation of the wave field by currents occurs at scales of 10–100 km, with $H_s$ varying by more than 50% on scales of 10 km. Romero et al. [10] presented a numerical study of CEW on the submesoscales (on the order of hundreds to tens of kilometers) with a realistic model configuration in Southern California. There it was identified that the modulation of wave field due to currents is larger for the wave-breaking variables (i.e., whitecap coverage, air-entrainment, and energy dissipation), followed by the resolved mean square slope, surface Stokes drift, and $H_s$. Background currents on average increased the directional spreading by 0.9° and modulated the mean wave direction within $\pm 5°$. Villas Bôas et al. [11] used the WaveWatch III (WWIII) third-generation numerical wave model to assess the relative effect of current divergence and vorticity in modifying several wave properties, including wave direction, period, directional spreading, and $H_s$. The authors found that the spatial variability of $H_s$ was highly sensitive to the nature of the underlying current, and that refraction is the primary mechanism that led to $H_s$ gradients. Marechal et al. [12] also used WWIII but studied the effect of wind waves within an eddy. There, they uncovered that wave amplitude, frequency, and direction were all sensitive to the presence of the underlying mesoscale structures which resulted from the destabilization of the eddy. Additionally, it was identified that the surface current vorticity and the intrinsic frequency of incident waves were key in the wave response to current modulation. Marechal and Ardhuin [13] showed that a phase-averaged numerical wave model forced with surface winds, realistic and high-resolution surface currents that was sufficiently directionally discretized could capture the sharp $H_s$ gradients observed by satellite altimeters.

To investigate CEWs within mesoscale eddies, this study selected an eddy in the South China Sea (SCS). As one of the largest semi-enclosed marginal seas of the Western North Pacific, the SCS is characterized by a highly complex current system and many thousands of ocean mesoscale eddies [14,15]. Consequently, a series of observations from the China–France Oceanographic SATellite (CFOSAT) and output from a third-generation numerical wave model, Simulating WAves Nearshore (SWAN), were used to catalogue and analyze how waves interact with mesoscale eddies. The remainder of this paper is structured as follows. The data and methodology are given in Section 2. Sensitivity analyses of the effects of an ideal mesoscale eddy on surface waves is provided in Section 3. Sections 4 and 5 provide results of an analysis of surface wave fields in the SCS from CFOSAT observations, and a summary of this study, respectively.

## 2. Data and Methodology

### 2.1. Data

2.1.1. Surface Wave and Wind CFOSAT Observations

To study wind and wave properties within a selected eddy, observations were derived from the Chinese-built wind scatterometer (SCAT), and French-built Surface Wave Investigation and Monitoring (SWIM) wave spectrometer onboard the recently launched (29 October 2018) CFOSAT platform. The swath widths of SCAT and SWIM are about 1000 km and 180 km, respectively, and provide simultaneous, co-located observations of surface wind and wave fields. The spatial resolutions of their wind and wave data are 12.5 km × 12.5 km and 70 km × 90 km, respectively [16]. SWIM is a Ku-band radar with a near-nadir scanning beam geometry to measure the spectral properties of surface ocean waves and can also provide estimates of $H_s$ along its track from nadir measurements every

~7 km [17,18]. SWIM wave spectra are characterized by a 180° ambiguity in wave propagation direction and the angular discretization is 15° [19]. Level 2a wind and wave products were utilized and the variables of $H_s$, wavelength (L), wave direction, and wind speed were acquired within the SCS on a geographical range from 5°N to 20°N, and 106°E to 116°E during the period from October 2019 to November 2019. The CFOSAT-SWIM data were collected from the AVISO+ website in France (https://www.aviso.altimetry.fr/home.html) (accessed on 1 December 2021) and National Satellite Ocean Application Service in China (https://osdds.nsoas.org.cn) (accessed on 1 December 2021).

### 2.1.2. AVISO Current and Sea Level Anomaly

To identify and track eddies, data from the Archiving, Validation, and Interpretation of Satellite Oceanographic (AVISO), multiple satellite-merged sea level anomaly (SLA) product was used in an eddy detection algorithm (Section 2.2.1). This dataset merges several satellite altimeter measurements to obtain a daily product, which has a spatial resolution of 0.25° × 0.25° and a daily temporal resolution. Data from 1 July 2019 to November 2019 were used in this study. Data can be acquired at http://www.aviso.oceanobs.com/ (accessed on 2 November 2021). AVISO geostrophic current anomaly data (vDT2018) with a spatial resolution of 0.25° × 0.25° and a daily temporal resolution were also used and can be obtained from https://cds.climate.copernicus.eu/cdsapp#!/dataset/satellite-sea-level-global?tab=overview (accessed on 2 November 2021). Alternatively, the same data can be acquired from the Global Ocean Mesoscale Eddy Atmospheric–Oceanic–Biological Interaction Observational Dataset (GOMEAD)(V1) at http://www.doi.org/10.11922/sciencedb.01190 (accessed on 2 November 2021) [20].

### 2.1.3. ERA5 Reanalysis

ERA5 is the fifth-generation reanalysis for the global climate and weather for the past four to seven decades and provides hourly estimates of a large number of atmospheric, ocean wave and land surface variables [21]. ERA5 reanalysis wave data with a spatial resolution of 0.5° × 0.5° and on an hourly temporal resolution were used and can be acquired at https://cds.climate.copernicus.eu/ (accessed on 1 October 2021). The mean wave direction was the mean direction of sea surface waves where the wave field consisted of a combination of waves of different heights, lengths, and directions. As such, this parameter was a mean over the entire two-dimensional wave spectrum, accounting for both wind waves and swell.

To eliminate ambiguity in CFOSAT-SWIM wave direction observations using ERA5 reanalysis data, a two-step procedure was conducted. First, 180° was added to the CFOSAT-SWIM wave direction observations, which gave rise to two numbers: (a) the original, ambiguous value, and (b) a new value which, due to the addition of 180°, eliminated the SWIM-inherent ambiguity in wave direction. Then, ERA5 wave direction from the closest geographical point to the point of CFOSAT-SWIM wave direction observations was acquired. This value was compared with the original, ambiguous value, and the new value (i.e., observation + 180°). Following this, whichever number ERA5 agreed with most, (at most it can only agree with one of them), then the 'true' value, within the limits of ERA5-inherent errors, was selected as the 'corrected' value. However, it is important to note that this technique should be used only in deep sea conditions where wave energy dissipation due to bathymetry is, negligible as publicly available datasets such as ERA5 may not necessarily be produced using high-resolution bathymetric datasets in their wave models, which would have a non-negligible effect on wave propagation directions.

### *2.2. Methodology*

### 2.2.1. Eddy Detection

To automatically detect and track mesoscale eddies, a vector geometry-based methodology developed by Nencioli et al. [22] was used. Using the geostrophic equilibrium relation, the AVISO sea level anomaly field was converted into a geostrophic velocity

field, and through identifying four geometric features mesoscale eddies could be detected. Though detailed in Nencioli et al. [22], briefly, the detection algorithm identified minima in the velocity fields, and once an approximate center in a prospective eddy was identified, boundaries were determined by four constraints. The eddy boundary was defined by the contour of the flow function. Due to weak divergences in the eddy velocity field, the velocity vector was tangent to the flow function contour and the flow tangential velocity increased along the normal direction. Consequently, the eddy boundary could be defined as the outermost closed contour around the identified central point. A global eddy dataset based on this algorithm has been published as the Global Ocean Mesoscale Eddy Atmospheric–Oceanic–Biological Interaction Observational Dataset (GOMEAD)(V1) [20]. Data can be acquired at http://www.doi.org/10.11922/sciencedb.01190 (accessed on 2 November 2021).

### 2.2.2. Research Scheme

Firstly, SWAN was used in a series of sensitivity experiments (i.e., different current velocity fields and wind speeds to generate waves) to study the influence of an idealized eddy in synthetic wave fields. Secondly, a typical eddy identified in the SCS and the distribution of wave characteristics over the eddy was examined using CFOSAT-SWIM wave track data (Figure 1) to validate the SWAN results. Using these results, a method to determine the footprint of eddies in oceanic wave fields was devised.

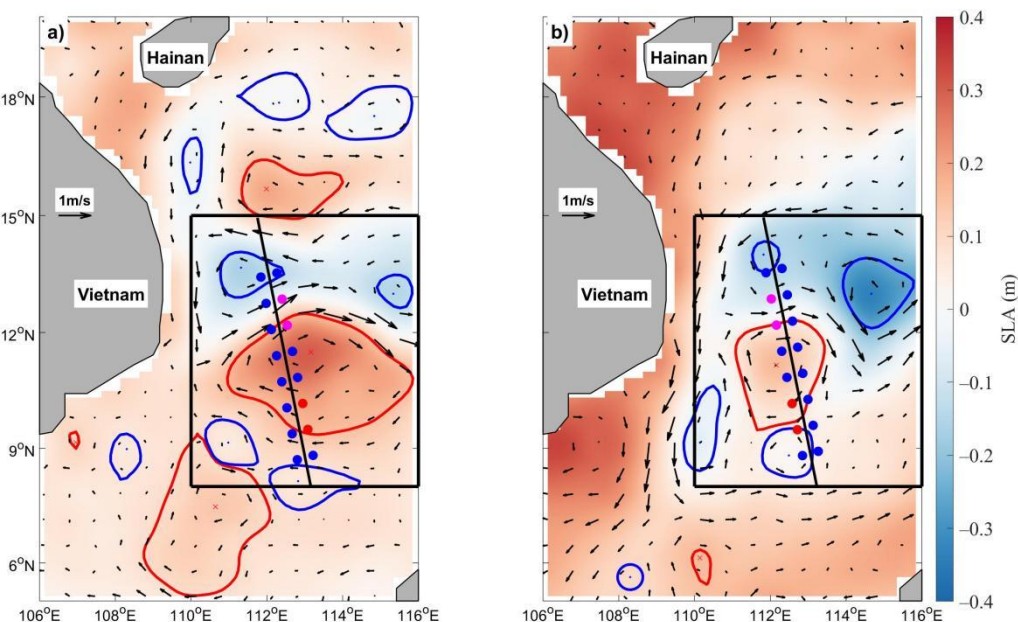

**Figure 1.** The current system and eddies in the SCS. The dots and black lines show CFOSAT-SWIM wave and wind track on (**a**) 8 October 2019, and (**b**) 16 November 2019. The shading color in the panels represent sea level anomaly (SLA). The black arrow represents AVISO geostrophic current anomaly. Red and blue lines demarcate warm- and cold-core eddies identified by the automatic eddy detection algorithm. In this study, attention is placed on the warm-core eddy in the black box (Section 4).

## 3. Sensitivity Experiments

In this section, sensitivity experiments of ideal mesoscale eddies on waves are carried out using SWAN. An eddy with different surface current velocities and a wave field forced by different wind speeds to generate waves of different heights are used to analyze the modulatory effect of eddies on waves.

### 3.1. Model Description

The SWAN model version 41.10 was used to simulate the wave characteristics in different cases. The SWAN model is a third-generation wave model based on the action

density balance equation, which is widely applied to simulate wave parameters in coastal areas, lakes, and estuaries [23,24]. The governing equation is as follows:

$$\frac{\partial N}{\partial t} + \nabla_{\vec{x}} \cdot \left[ \left( \vec{c}_g + \vec{U} \right) N \right] + \frac{\partial c_\sigma N}{\partial \sigma} + \frac{\partial c_\theta N}{\partial \theta} = \frac{S_{tot}}{\sigma} \tag{1}$$

where $N\left(\vec{x}, t; \sigma, \theta\right)$ is the evolution of the action density in space $(\vec{x})$, time t, frequency ($\sigma$) and propagation direction ($\theta$). N is defined as $N = E(k)/\sigma$ and is conserved during propagation in the presence of ambient current, whereas energy density E is not. The quantities $c_\sigma$ and $c_\theta$ are the propagation velocities in spectral space $(\sigma, \theta)$. The ambient current is denoted as $\vec{U}$. $S_{tot}$ on the right side of the equation represents the source function term that controls physical processes. The source term $S_{tot}$ is expressed as:

$$S_{tot} = S_{in} + S_{nl3} + S_{nl4} + S_{ds,w} + S_{ds,b} + S_{ds,br} \tag{2}$$

where $S_{in}$ is the wind energy input, the nonlinear wave–wave interaction includes three-wave interactions ($S_{nl3}$) and four-wave interactions ($S_{nl4}$), which play a major role in shallow and deep water, respectively. The dissipation terms include whitecapping ($S_{ds,w}$), bottom friction ($S_{ds,b}$), and depth-induced wave breaking ($S_{ds,br}$). A detailed description concerning SWAN can be found in the official manual available at http://swanmodel.sourceforge.net/ (accessed on 1 June 2021). In this study, wind energy input ($S_{in}$), four-wave interactions ($S_{nl4}$), and whitecapping ($S_{ds,w}$) were included for simulations. Three-wave interactions ($S_{nl3}$), bottom friction ($S_{ds,b}$), and depth-induced wave breaking ($S_{ds,br}$) were not considered.

### 3.2. Model Configuration

According to wave propagation theory, waves are dispersive in deep water and propagate with currents at the group velocity [12]. CEW depends on the ratio of wave group velocity ($c_g$) to current velocity. Therefore, to study the factors affecting the CEW and how CEW contributes to wave evolution, sensitivity analyses of the eddy current velocity and overlying wind speed were carried out by using a set of experiments (Table 1). The maximum eddy surface current velocities (U) were set to 0.2, 0.5, 0.8, and 1.0 m/s. The wind speed (W) was set to 5, 10, 15, 20 m/s. The initial spectrum peak periods were 2.23, 3.71, 5.02, and 6.13 s under the various wind speeds without currents. The corresponding wave phase velocities ($c_p$) were 3.38, 5.79, 7.83, 9.56 m/s. Changes in the wave age ($c_p/W$) and the ratio of wave group to current velocity ($c_g/U$) are discussed.

**Table 1.** Experimental setup.

| Case Number | Current Velocity (U; m/s) | Wind Speed (W; m/s) | Peak Period (Tp; s) | Wave Phase Velocity ($c_p$; m/s) | Wave Age ($c_p/W$) | $c_g/U$ |
|---|---|---|---|---|---|---|
| 1 | 0.2 | 5 | | | | 8.45 |
| 2 | 0.5 | 5 | | | | 3.38 |
| 3 | 0.8 | 5 | 2.23 | 3.38 | 0.696 | 2.11 |
| 4 | 1.0 | 5 | | | | 1.69 |
| 5 | 0.2 | 10 | | | | 14.48 |
| 6 | 0.5 | 10 | | | | 5.79 |
| 7 | 0.8 | 10 | 3.71 | 5.79 | 0.579 | 3.62 |
| 8 | 1.0 | 10 | | | | 2.90 |
| 9 | 0.2 | 15 | | | | 19.58 |
| 10 | 0.5 | 15 | | | | 7.83 |
| 11 | 0.8 | 15 | 5.02 | 7.83 | 0.552 | 4.89 |
| 12 | 1.0 | 15 | | | | 3.92 |
| 13 | 0.2 | 20 | | | | 23.90 |
| 14 | 0.5 | 20 | | | | 9.56 |
| 15 | 0.8 | 20 | 6.13 | 9.56 | 0.478 | 5.98 |
| 16 | 1.0 | 20 | | | | 4.78 |

The SWAN wave model was run in nonstationary mode. By default, the initial condition is the JONSWAP spectrum with a $\cos^2(\theta)$ directional distribution centered around the local wind direction. As previously described, only wind energy input ($S_{in}$), four-wave interaction ($S_{nl4}$), and whitecapping ($S_{ds,w}$) were considered in the simulation. Four-wave interactions were estimated by the Discrete Interaction Approximation (DIA) where $\lambda = 0.25$ and $Cnl4 = 3 \times 10^7$ [25]. The whitecapping expression by Janssen was used [26,27]. Partially modeled diffraction was added to the model using a phase-decoupled refraction diffraction approach [28].

In all tests, the eddy position was set in the middle of the region. The ideal center velocity and peripheral velocity of eddy were both 0 m/s. Let the point denoted by $x_0$ and $y_0$ be the eddy center. r is the distance from any point to the eddy center:

$$r = \sqrt{(x_i - x_0)^2 + (y_i - y_0)^2} \qquad (3)$$

Setting $lr = \begin{cases} r, \left(r < \frac{r_0}{2}\right) \\ r_0 - r, \left(\frac{r_0}{2} \leq r \leq r_0\right) \\ 0, (r > r_0) \end{cases}$ , $r_0$ is eddy radius. In the simulation, $r_0 = 20,000$.

The eddy surface velocity is given by:

$$\begin{cases} u_x = A \times lr \times \frac{(x_i - x_0)}{r_0} \\ u_y = -A \times lr \times \frac{(y_i - y_0)}{r_0} \end{cases} \qquad (4)$$

The eddy surface current velocity varies with the distance from the eddy center. The current velocity within $\frac{r_0}{2}$ from the eddy center increases continuously and decreases from $\frac{r_0}{2}$ to $r_0$. The maximum velocity is at the location of $\frac{r_0}{2}$. The velocity outside $r_0$ is 0 m/s. The maximum velocity is adjusted by changing parameter A. In the experiments, a clockwise-rotating anticyclonic eddy in the northern hemisphere was used as an example. The eddy radius was set to 20 km and wind direction was from west to east. The eddy velocity distribution estimated by Equation (4) is provided schematically in Figure 2.

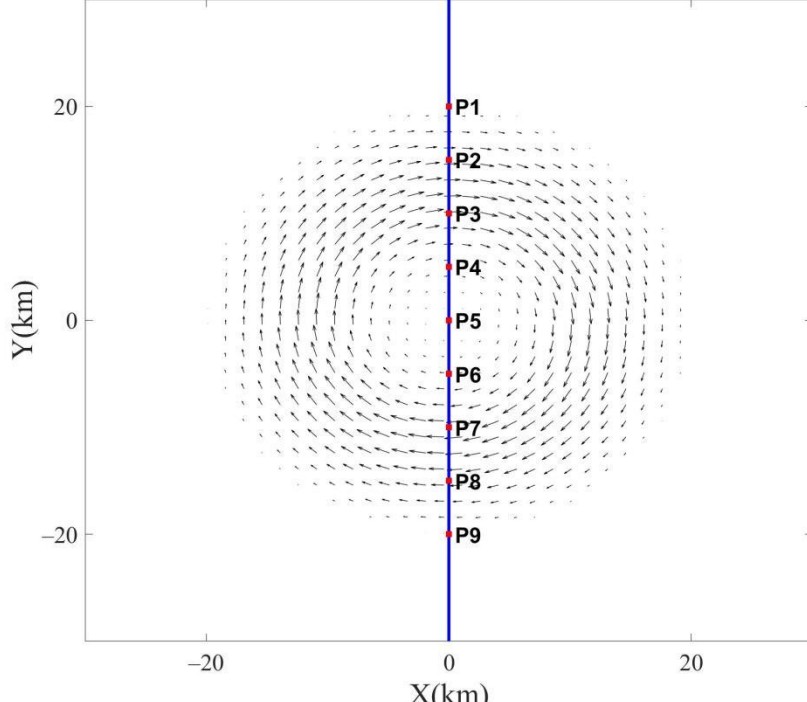

**Figure 2.** Eddy surface current velocity distribution diagram. Black arrows represent current vectors. The blue line and red dots are the section used to analyze the wave parameter variations in Section 3.3.

The parameter settings of the SWAN model were as follows. Calculations were performed on a Cartesian grid of 100 km × 100 km. The grid spatial resolution was 100 m × 100 m. The number of grid points was 1000 × 1000. The model was run for two days with a time step of 2 min. The directional wave energy density spectrum function was discretized using 24 directional bins and 32 frequency bins between 0.0418 Hz and 1.0 Hz. Other parameter settings are shown in Table 2. In each test, wave trains propagated from the western boundary, generated by western, constant winds at the various wind speeds. The water depth (bathymetry) was set to 1000 m in all cases. To analyze the effect of eddies, model runs were performed with and without currents added.

**Table 2.** SWAN model parameter settings.

| Parameter | Value |
| --- | --- |
| Time Step (min) | 2 |
| Number of directional bands | 24 |
| Number of frequencies | 32 |
| Discrete frequency range (Hz) | 0.0418–1 |
| Physical process | JANSSEN |
| Spectral pattern | JONSWAP |
| Propagation scheme | Backward Space Backward Time (BSBT) |
| Spatial resolution | 100 m × 100 m |
| Temporal resolution (min) | 2 |
| Type of spectral model | Deep water |
| Coordinate Mode | Cartesian Nonstationary 2D mode |

### 3.3. Wave Field Variability in an Idealized Eddy

The variability of incident waves generated by different wind speeds following propagation into eddies with different current velocities is investigated in this section. Waves were dispersive in deep water and were propagating in the current at the group velocity ($C_g$). CEW depended on the ratio of wave group velocity to current velocity. Surface currents modulated wave fields in terms of $H_s$, peak wave direction ($\theta_w$), mean wave period (T), and average wave steepness ($\delta$, the ratio of $H_s$ to L). The response of these wave parameters under different eddies are described below.

### 3.3.1. Significant Wave Height

Surface currents induce strong variability in $H_s$, and this is especially significant in highly solenoidal current fields such as those found in eddies [13]. Results for changes in $H_s$ under the sixteen cases (Table 1) are presented in Figure 3. Model runs without currents added are also given for ease of comparison. Shadings given relative differences are estimated as the subtraction of modeled results without currents from those with currents. Relative differences are the ratios of the differences to the results without current. The most readily observable characteristic was the asymmetrical distribution of wave variability (Figure 3a–d). A dipole-like structure formed where in the upper region demarcated by Y = 50–70 km $H_s$ decreased but was accompanied by a simultaneous increase in the lower region (Y = 30–50 km). This occurred because the angle between waves and currents (WCA) was less than 90° in the upper region, but larger than 90° in the lower region. This resulted in waves following the current (upper region) but opposing the current (lower region). CEW became more apparent as current velocities increased (Figure 3d,h,l,p). Interestingly, when the wind speed decreased (Figure 3a–d), CEW also became strong. This was due to the ratio of wave group velocity to the current velocity. The closer the ratio was to 1, the stronger the CEW and vice versa. Generally, the wave group velocity was much larger than the current velocity. When wind speeds were low, and wave heights were small; the closer the wave and current velocities were to one another, the stronger the CEWs (Table 1, column 7). In column 6 of Table 1, and Figure 3a–d, it can be seen that the higher the wave age (and hence, the more mature the wave age), the stronger CEW was. In Case 4 when W was 5 m/s and U was 1 m/s, CEW reached its maximum value of 20%. It can also

be observed in Figure 3c,d,h,l that $H_s$ increased in the outer age of the upper area where CEW values were negative (X = 50–70 km, Y = 50–70 km). $H_s$ decreased in the lower area (X = 50–70 km, Y = 30–50 km). This result is consistent with Marechal et al.'s [12] result.

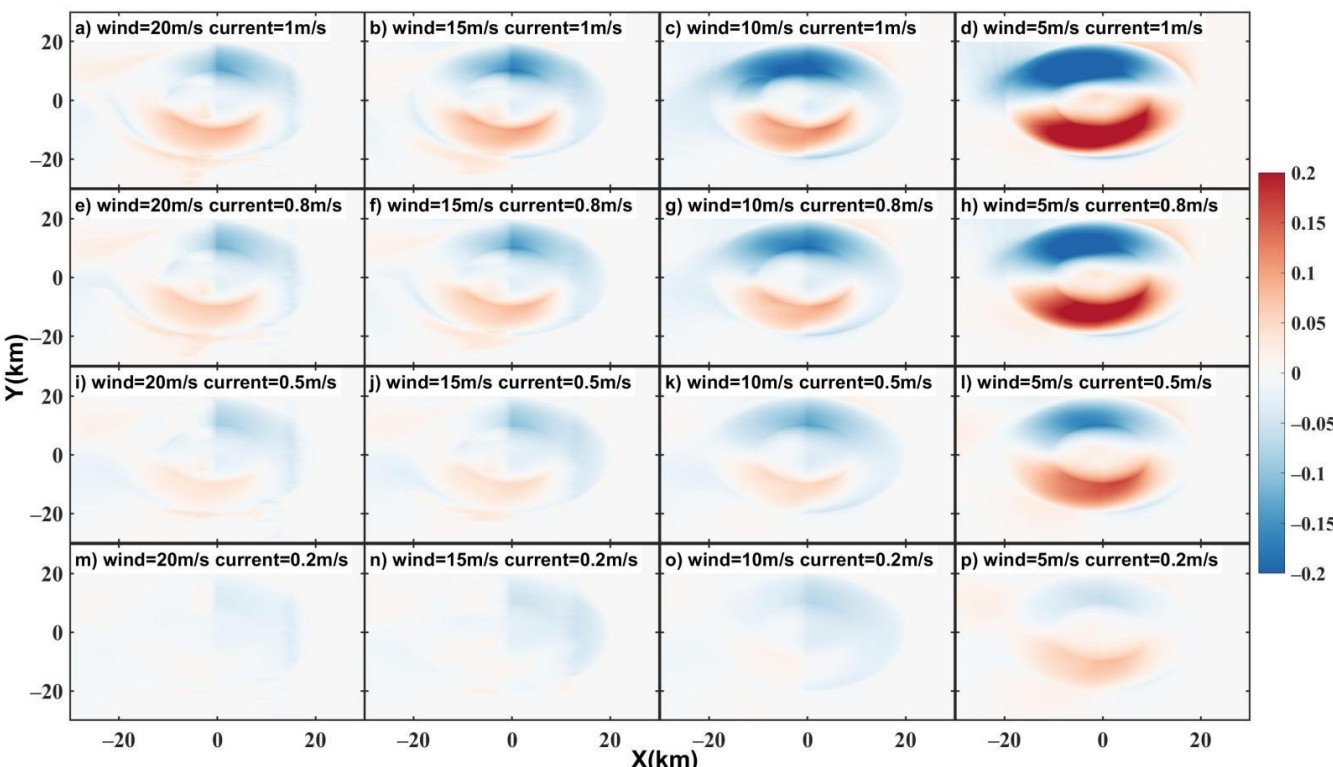

**Figure 3.** Relative $H_s$ differences modeled under the 16 cases listed in Table 2. The background color in the panels represents the relative differences.

### 3.3.2. Mean Wave Period

The distribution of T variability (Figure 4) was nearly opposite of $H_s$ (Figure 3). That is, inverse to the $H_s$ case, T increased in the upper part of the eddy (Y = 50–70 km), but decreased in the lower part (Y = 30–50 km). When wave and currents propagated/flowed in the same direction, both the wave velocity and L increased, leading to increased T and lower $H_s$. In counter-current regions, both wave velocity and L decreased, leading to smaller T and larger $H_s$. Variability in T was also influenced by $c_p/W$ and $c_g/U$. When waves were mature (i.e., higher wave ages; Column 6, Table 1), the relative differences in T were more readily observable (Figure 4a–d). When Cp/W = 0.696 and Cg/U = 1.69 (Case 4), T was influenced by CEW the most (more than 20%).

### 3.3.3. Average Wave Steepness

Figure 5 shows snapshots of the relative differences for relative δ differences. It was found that the distribution of δ variation was similar to $H_s$ (Figure 3). When waves and current propagated against one another (Y = 30–50 km), $H_s$ increased and T decreased (Figure 4), which led to a shortened L. Therefore, δ increased. Similarly, the wave steepness variation decreased at the upper part (Y = 50–70 km). Accordingly, when waves and current propagated against one another, $H_s$ became higher and waves became steeper, which led to wave focusing. When waves and current propagated/flowed in the same direction, $H_s$ became smaller and eventually flattened, leading to wave defocusing. Currents modified the δ most strongly when wave age was larger, and the wave group velocity was close to current velocity (Case 4).

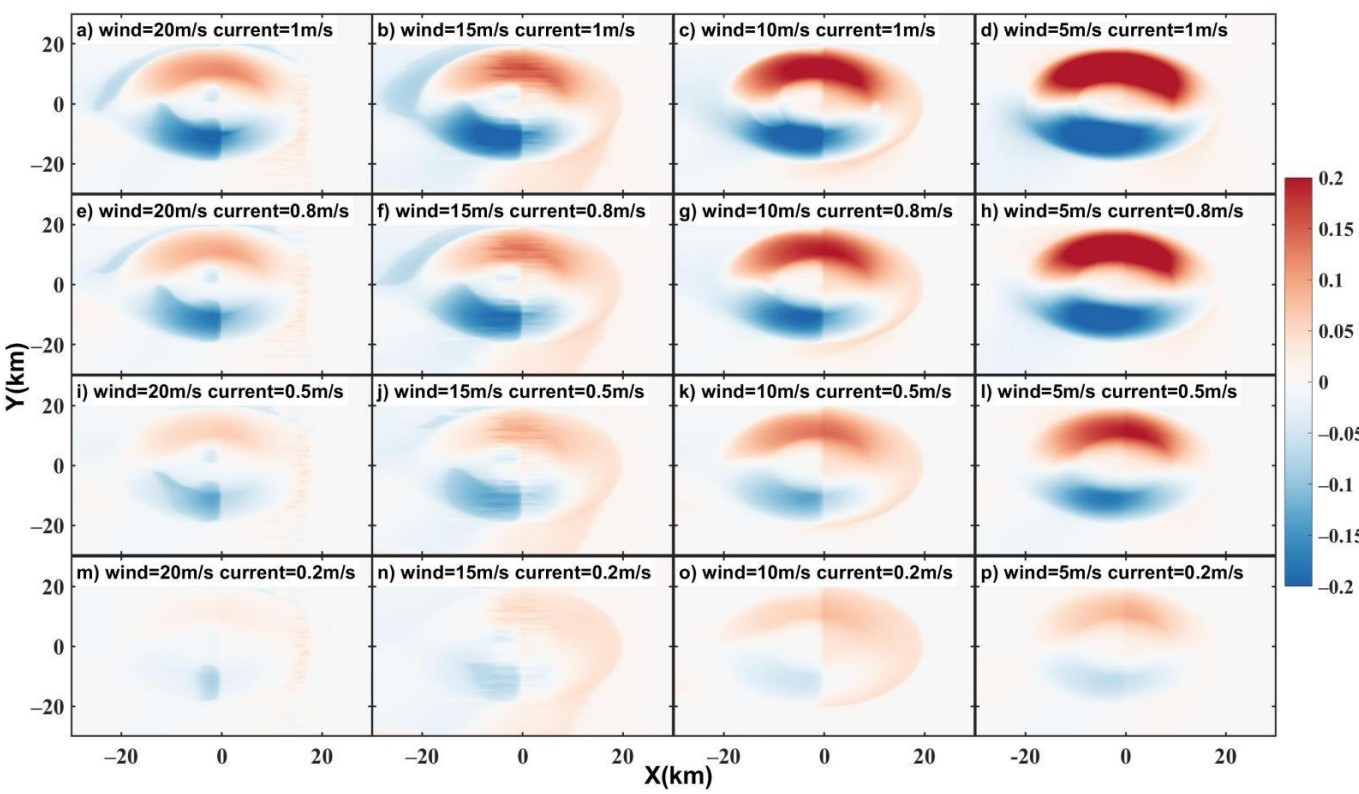

**Figure 4.** Same as Figure 3, but for peak wave period.

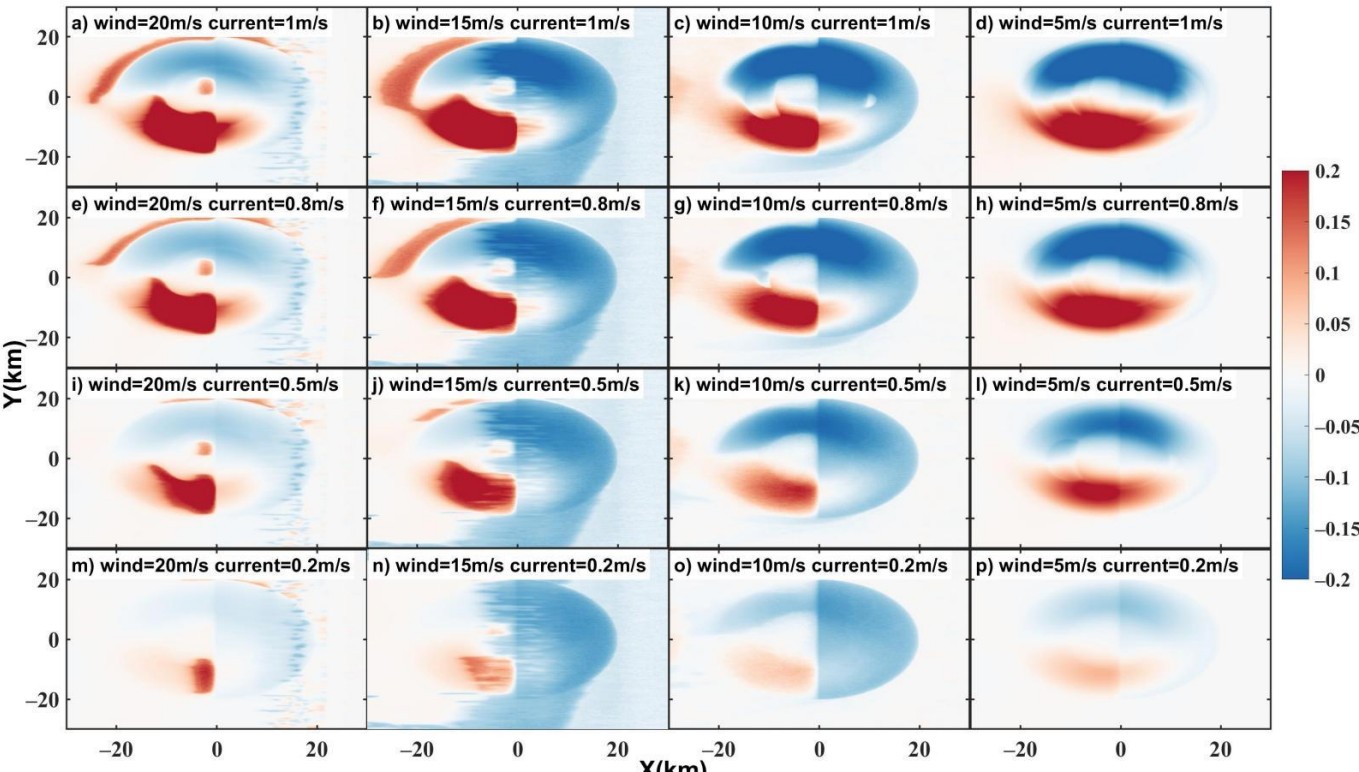

**Figure 5.** Same as Figure 3, but for wave steepness.

### 3.3.4. Peak Wave Direction

In each experiment, waves propagated west to east with an initial wave direction of 90° (0° is geographical north, rotating clockwise). The refraction induced by the surface currents could be captured by changes $\theta_w$. Due to refraction, waves turned within the current fields. In Figure 6, it can be seen that at the upper part (WCA < 90°) of the eddy, the wave direction deflected northward towards the outer edge (i.e., Y = 60–70 km) and the wave direction angle decreased. In the inner area (i.e., Y = 50–60 km), an opposite change occurred. The wave direction was then southward. The different wave direction variation between Y = 60–70 km and Y = 50–60 km caused wave defocusing.

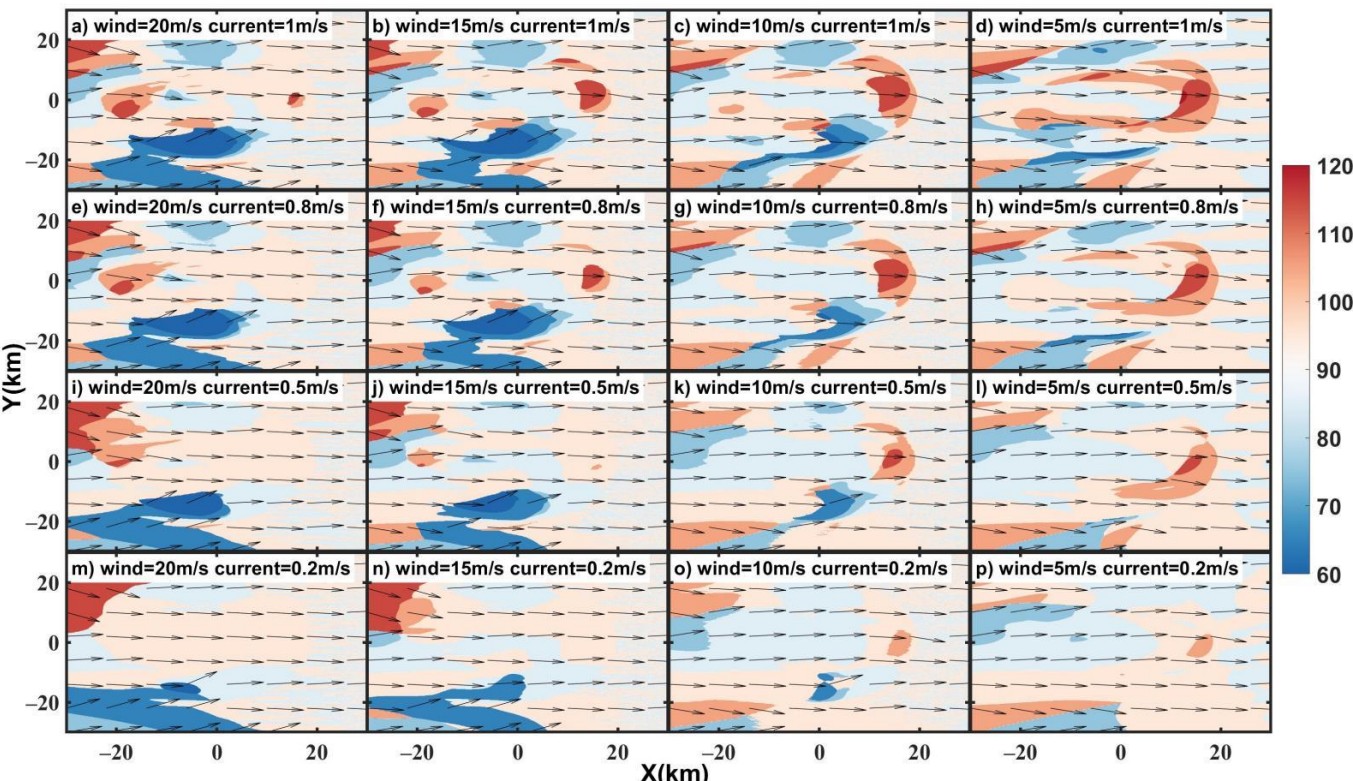

**Figure 6.** Instantaneous peak wave direction ($\theta_w$) distribution. The background color represents the peak wave direction values. Arrows are the peak wave direction.

At the lower part (WCA > 90°), waves propagated northward and the wave direction angle decreased (Y = 20–40 km). This was especially the case at the eddy edge (Y = 30 km), which resulted in wave focusing. This trend was stronger when currents were faster. Changes in wave direction formed a dipole pattern. In addition, at the rear of the eddy tail (X = 60–70 km, Y = 50 km), the wave obviously propagated southward and the wave direction angle increased, just like the wake vortex. This trend was strongest when the CEW was strongest (Case 4). The variability in wave direction was almost 30°. Moreover, the larger the wave age, the more significant the changes in wave direction became (Figure 6a–d). Mature waves were influenced by the CEW more strongly.

### 3.3.5. Statistical Analysis

In order to analyze the proportion of the CEW to wave distribution, the maximum percentage variation of wave parameters under 16 cases is listed in Table 3. Moreover, the maximum variation of simulated wave fields under 16 cases are plotted in Figure 7. As shown in Figure 7, $H_S$, $\delta$, T, and $\theta_w$ all increased with W and U. It can be found in Table 3 that the percentage variation of wave parameters caused by the CEW was the largest when the wind speed was 5 m/s and maximum eddy surface current velocity

was 1 m/s (Case 4). In other words, Case 4 had the largest influence of CEW in all the experiments. The maximum percentage variability of $H_S$, $\delta$, $T$, $\theta_w$ reached 30.31%, 35.02%, 41.01%, and 40°, respectively. Case 13 had the smallest percentage variation of the wave parameters. The corresponding variations of $H_S$, $\delta$, $T$, $\theta_w$ were 4.90%, 11.03%, 3.55%, and 30°, respectively. This phenomenon reflected the CEW, which was mainly related to the ratio of wave group velocity to current velocity. The closer the two velocities, the larger the wave age, the greater the CEW, and vice versa.

**Table 3.** Variations of wave parameters at different cases.

| Case Number | $\frac{\Delta H_s}{H_s}$ | $\frac{\Delta \delta}{\delta}$ | $\frac{\Delta T}{T}$ | $\Delta \theta_w$ (°) |
|:---:|:---:|:---:|:---:|:---:|
| 1 | 7.75% | 10.23% | 9.40% | 20 |
| 2 | 17.39% | 20.47% | 20.57% | 30 |
| 3 | 26.61% | 29.64% | 32.47% | 40 |
| 4 | 30.31% | 35.02% | 41.01% | 40 |
| 5 | 7.44% | 14.76% | 8.26% | 30 |
| 6 | 13.93% | 20.15% | 14.41% | 30 |
| 7 | 20.12% | 25.15% | 20.29% | 40 |
| 8 | 23.87% | 28.28% | 24.21% | 40 |
| 9 | 9.45% | 12.11% | 5.25% | 30 |
| 10 | 14.04% | 12.08% | 9.56% | 40 |
| 11 | 18.46% | 14.38% | 13.70% | 40 |
| 12 | 21.39% | 21.18% | 16.37% | 40 |
| 13 | 4.90% | 11.03% | 3.55% | 30 |
| 14 | 8.78% | 12.59% | 7.50% | 40 |
| 15 | 12.06% | 17.99% | 13.35% | 40 |
| 16 | 13.16% | 34.05% | 14.08% | 40 |

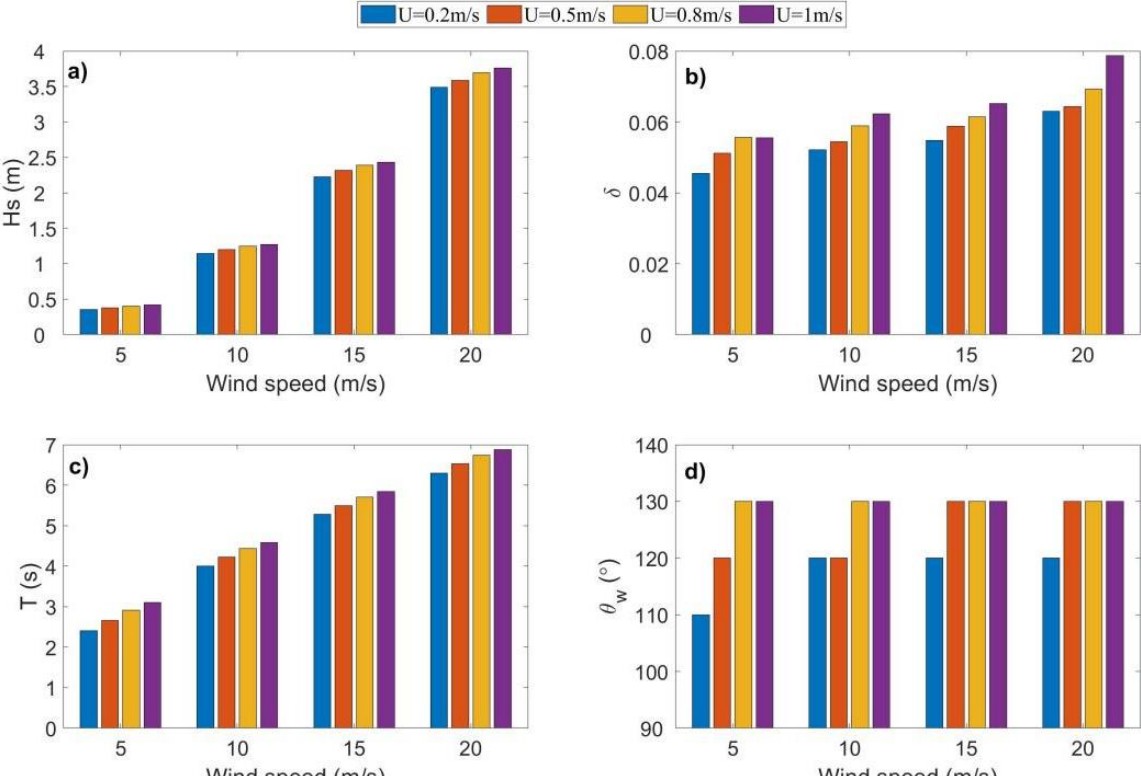

**Figure 7.** Wave parameters (**a**) $H_S$, (**b**) $\delta$, (**c**) $T$, and (**d**) $\theta_w$ for the 16 cases in Table 1. The horizontal coordinates represent the different wind speeds (5, 10, 15, and 20 m/s).

In order to show the CEW to wave distribution more intuitively, the wave parameters of each point under Case 16 (Figure 2; P1–P9) are drawn in Figure 8. It can be seen that from P1 to P9, the wave height and average wave steepness reached the minimum at P3 (corresponding to the upper part of Figures 3a and 5a) and the maximum at P7 (corresponding to the lower part of Figures 3a and 5a), while the mean wave period changed in reverse. The peak wave direction became larger at P3 and smaller at P7, which corresponds to Figure 6a. The analysis of wave distribution from the perspective of this section can deepen the understanding of wave field variations.

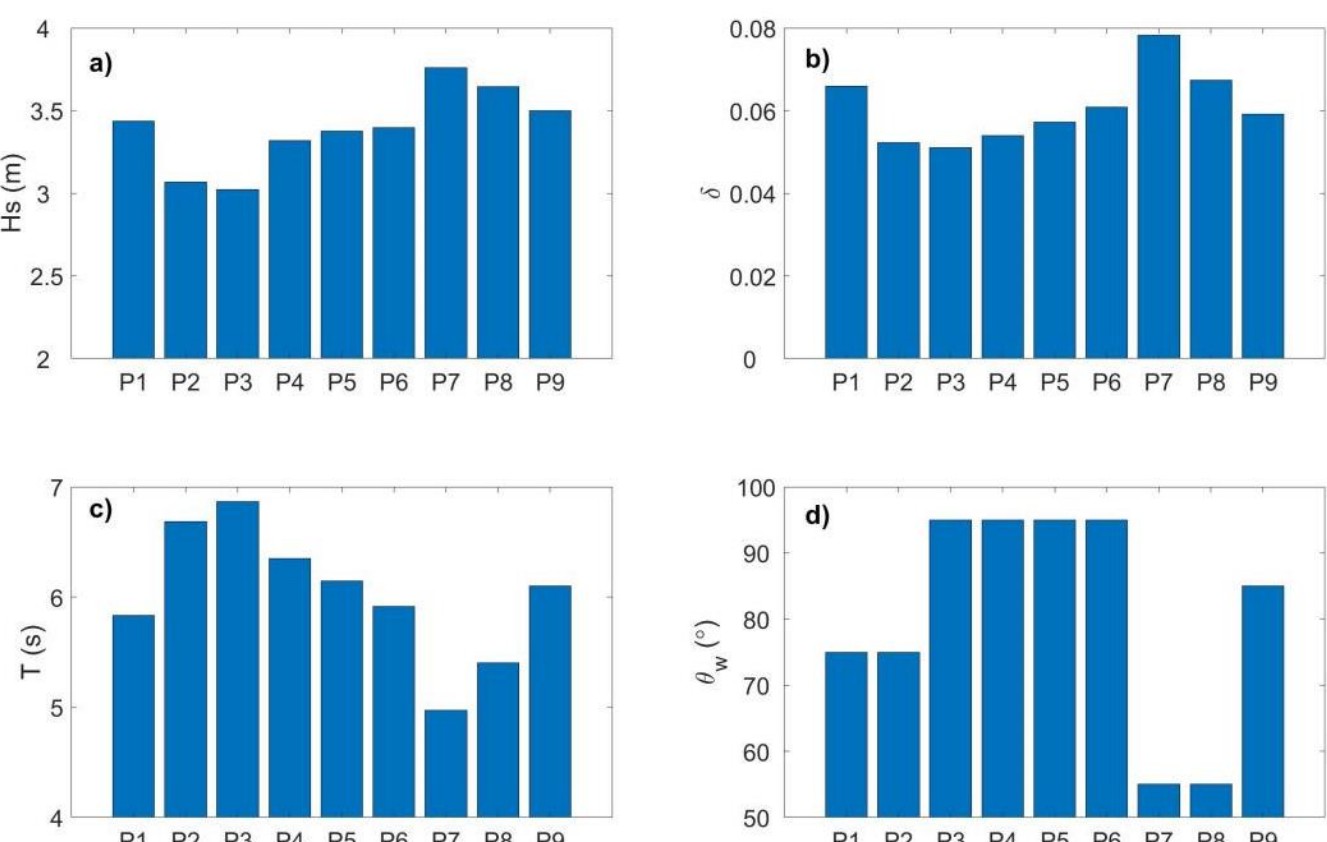

**Figure 8.** Wave parameters (**a**) $H_S$, (**b**) $\delta$, (**c**) T, and (**d**) $\theta_w$ along the section shown in Figure 2 under Case 16.

*3.4. Spectral Analysis*

In order to quantify the energy transport from ocean currents to waves, Ardhuin et al. [2] used numerical models to simulate waves and currents in the Gulf Stream. There it was identified that the spectra of $H_s$ and U showed that open ocean currents (eddies, fronts, and filaments) could be the main source of the variability in $H_S$ at scales of 10–100 km. To study the energy transport between eddy and waves, spectral analysis was applied to $H_s$ in all cases. The specific steps are described by Wang et al. [29]. As shown in Figure 9, wave spectra were quite different under different wind fields. When the wind speed was 5 m/s, the wave spectra under different current velocities had large differences, that is, U had a great influence on the wave energy distribution. When the wind speed was 20 m/s, the five spectra under different eddy surface current with 0, 0.2, 0.5, 0.8, and 1 m/s had little difference, which meant that the wave energy distribution was minutely affected by eddies. From Figure 9a, it can be seen that the higher the current velocity, the higher the wave energy. The wave energy simulated with eddy's maximum current velocity of 1 m/s was highest, followed by the simulation with eddy's maximum current velocity of 0.8, 0.5, and 0.2 m/s. The minimum wave energy was estimated by the wave field simulated without current. The wave energy distribution caused by the eddy current mostly occurred in the

spatial scale of 5–90 km, which belonged to the submesoscale and mesoscale eddies. The last oscillation when k ~ 0.5–8 km$^{-1}$ was probably related to the numerical dissipation.

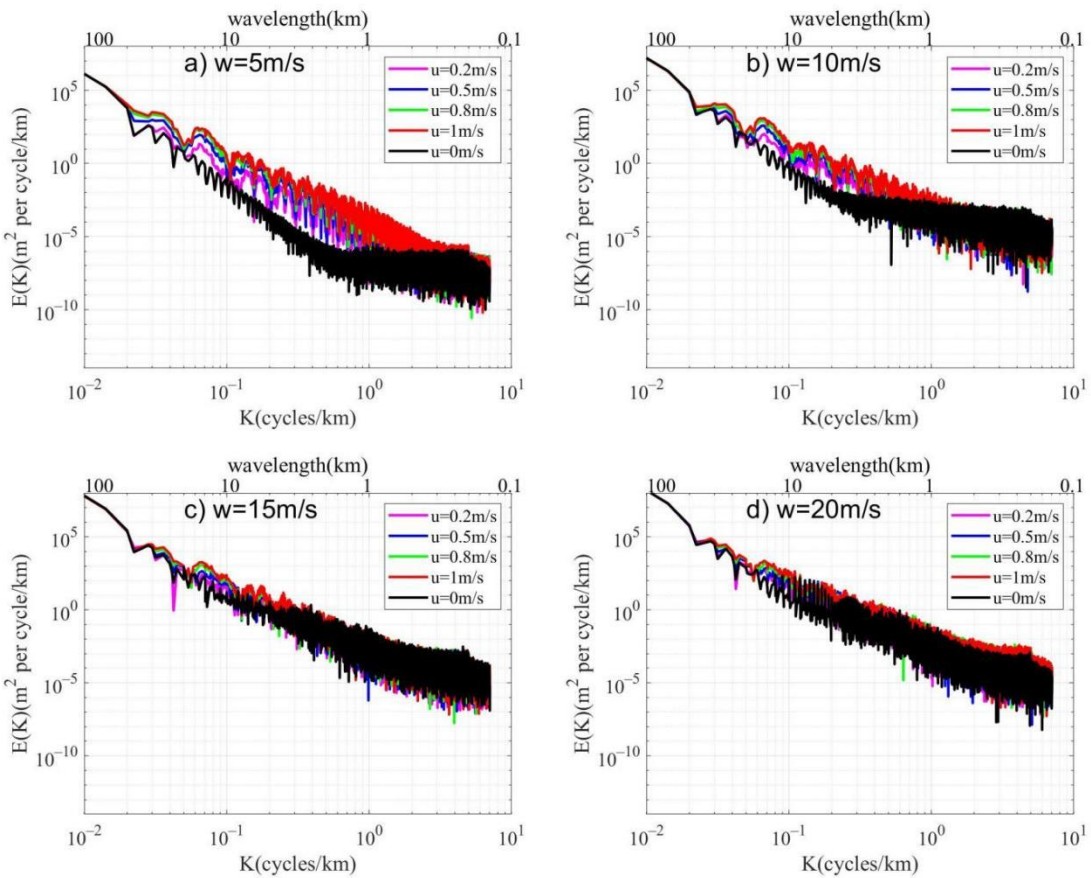

**Figure 9.** 1D spectra of significant wave heights (H$_s$) under wind speeds of (**a**) 5 m/s, (**b**) 10 m/s, (**c**) 15 m/s, and (**d**) 20 m/s with without currents (black lines) and with current velocities of 0.2 (magenta), 0.5 (blue), 0.8 (green), and 1 m/s (red), respectively.

## 4. Observation of the Effect of Mesoscale Eddies on Surface Waves in the SCS

### 4.1. Two Examples of the Effect of Mesoscale Eddies on Surface Waves in the SCS

In this section, a large warm eddy located at 9–13°N in the SCS was selected as an example to find the footprint of the mesoscale eddy on wave distribution. Figures 10 and 11 represent the examples on 8 October 2019, and 16 November 2019, respectively. Figures 10e and 11e are the zoomed black boxes of Figure 1a,b. The red line represents a warm-core eddy, which is focused on. The dots A1–A8 are the CFOSAT-SWIM wave track. The H$_s$, δ, WCA, and U of these points are plotted in Figures 10a–d and 11a–d. Among them, H$_S$ is the observation data, and δ is the ratio of the CFOSAT-SWIM-observed H$_S$ to L. Here, WCA is the angle between SWIM wave direction and current direction. The ambiguous CFOSAT-SWIM-observed wave direction was corrected by ERA5 reanalysis. The current velocity used was the AVISO current data interpolated onto the wave track (A1–A8).

As shown in Figures 10a–b and 11a–b, both significant wave height and wave steepness were significantly higher at A6 and A7 than at A2 and A3. According to Figures 10e and 11e, the northeast monsoon prevailed in autumn in the SCS, which meant that the WCA at 12–13°N (A6 and A7) was 90°–180° (wave and current in opposite direction) and 0°–90° (wave and current in the same direction) at 9–10°N (A2 and A3). A6 and A7 were in the area which was equivalent to the lower part of Figures 3 and 5. By contrast, A2 and A3 were in the area which was equivalent to the upper parts of Figures 3 and 5. This phenomenon verifies the conclusion of Section 3. Moreover, it can be deduced from Section 3 that, under the same wave phase velocity, the larger the current velocity, the greater the CEW.

Observational data was useful to understand this point. A6 and A7 with larger current velocity (shown in Figures 10d and 11d) caused larger wave variations than A2 and A3. In addition, the difference of $H_s$ between the two sides (A6, A7 and A2, A3) of the eddy was about 0.6–0.7 m, which accounted for about 20–30%. This conclusion is consistent with Section 3, and verifies the accuracy of the model results.

### 4.2. Discussion on the Effect of Wind Speed on Wave Distribution

Winds and currents are the important factors that affect waves. Firstly, the wind speed in the eddy area should be discussed. Previous studies have shown that eddies can affect sea surface wind speeds [30]. Secondly, while the variability caused by currents should also be analyzed, a separate analysis of these two factors requires the estimation of wind waves by an empirical formula and then to subtract that value from the $H_s$. The difference between these two quantities would be the CEW. Here, wind speed from CFOSAT tracks was analyzed so that the influence of winds could be excluded and the influence of CEW clearly identified. Figure 12 shows a 10 m wind speed distribution over warm-core eddies on 16 November and 8 October 2019. On the two sides of the eddies, the wind speed near 12°N was not significantly higher than the wind speed near 10°N. The wind speed was also small, especially on 16 November 2019 (Figure 12a). This excludes the wave height increase at A6 and A7 caused by wind speed. Therefore, the wind speed was not the main reason for differences in wave activity on both sides of the eddy on the two days. Instead, the CEW was the primary factor governing wave variability on both sides of the eddy in the two examples.

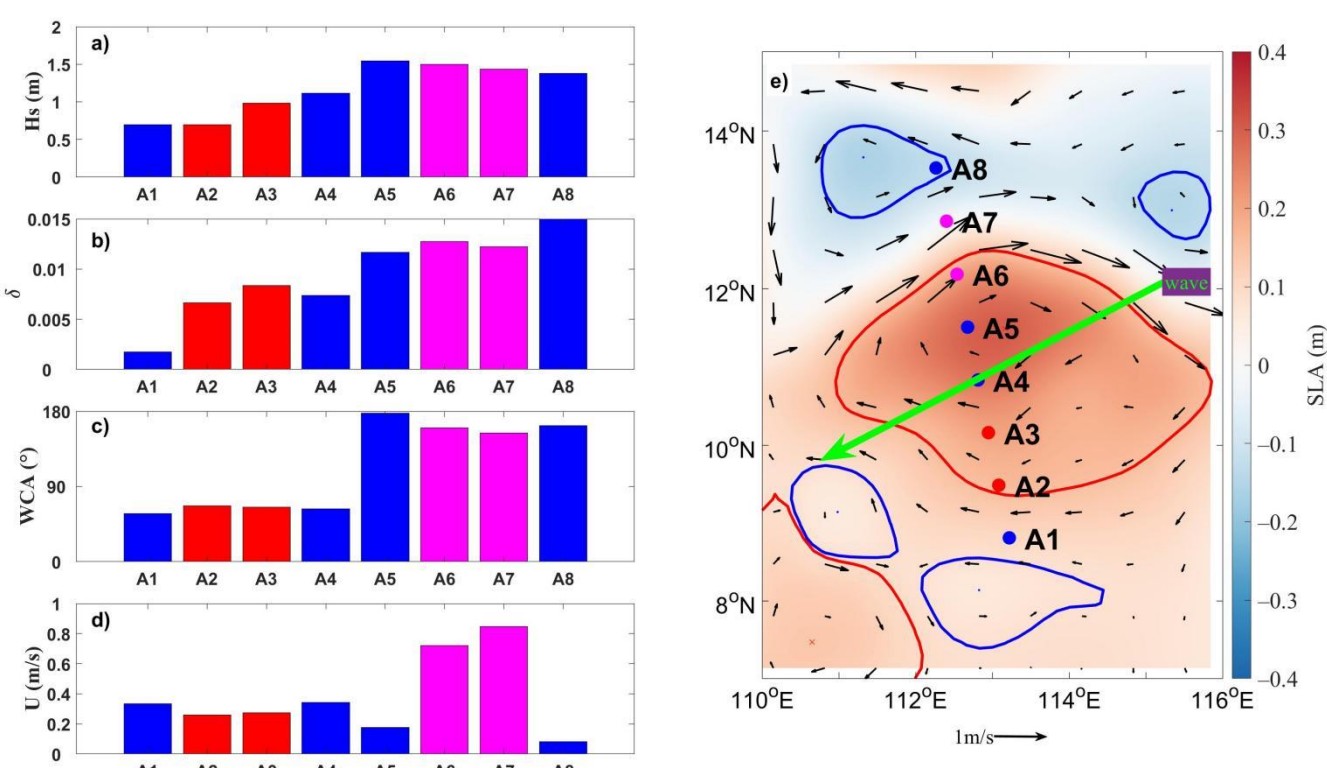

**Figure 10.** Variation of (**a**) $H_s$, (**b**) $\delta$, (**c**) WCA, and (**d**) U on the CFOSAT-SWIM wave track. Panel (**e**) is a zoomed black box of Figure 1. The dots in (**e**) represent the CFOSAT-SWIM wave track on 8 October 2019. The two red dots represent wave and currents propagating/flowing in the same direction. The two magenta dots represent the waves and currents propagating/flowing against one another. The green arrow represents the wave direction of ERA5 data in the area. The shading color, the black arrow, the red and blue lines are the same as in Figure 1.

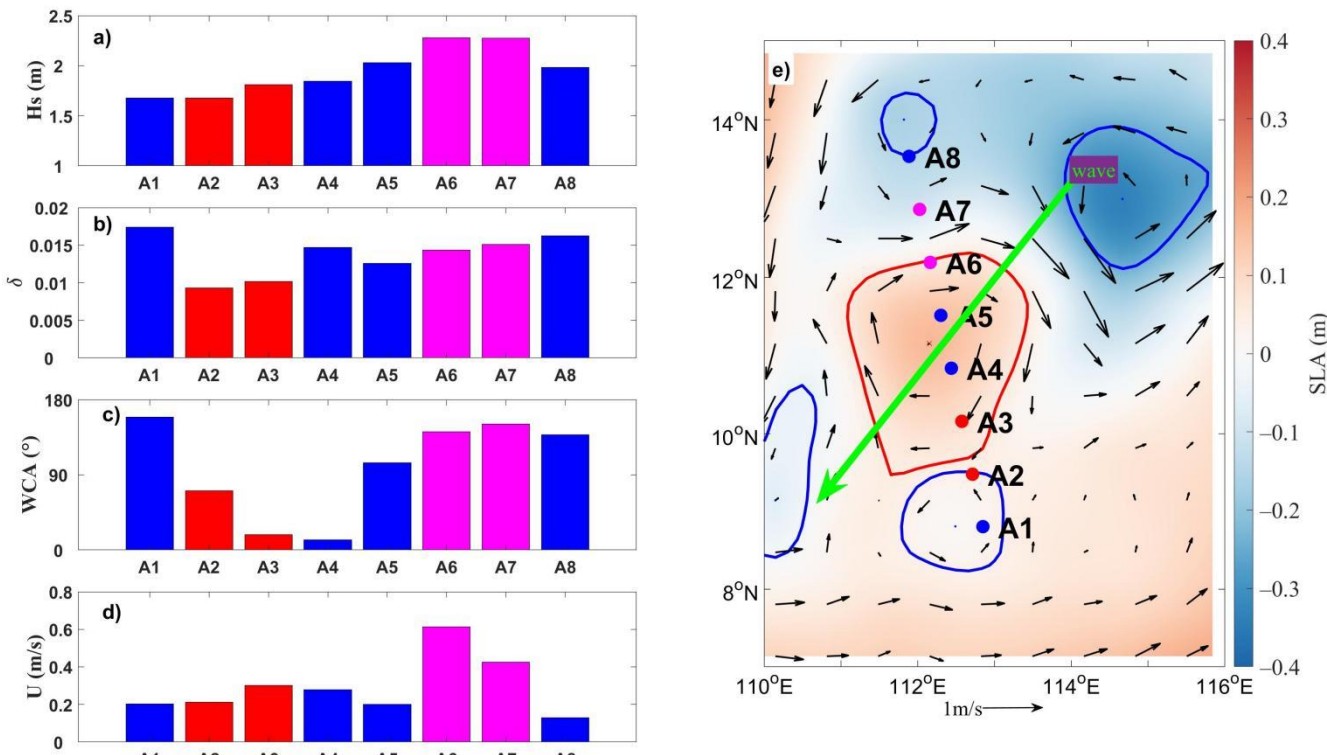

**Figure 11.** Variation of (**a**) H$_s$, (**b**) δ, (**c**) WCA, and (**d**) U on the CFOSAT-SWIM wave track. Panel (**e**) is a zoomed black box of Figure 1. The variables are defined as Figure 10, but for 16 November 2019.

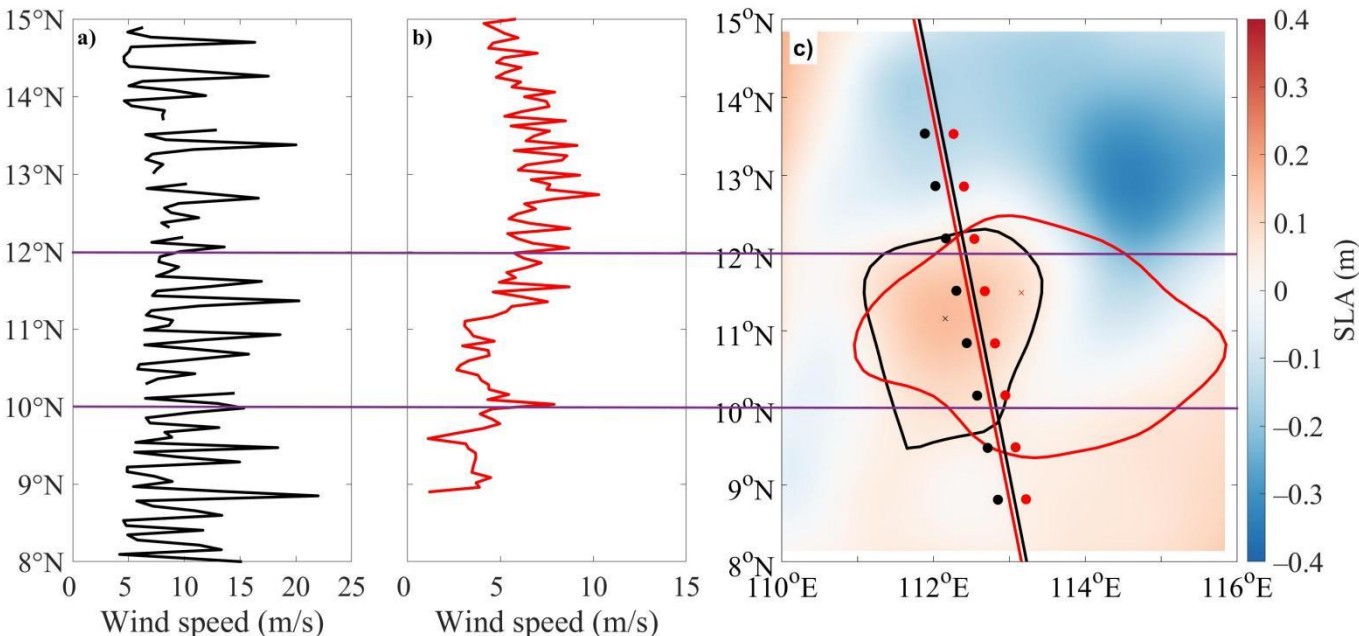

**Figure 12.** The wind distribution of the CFOSAT-SWIM wind tracks on (**a**) 16 November 2019, (**b**) October 2019, and (**c**) eddy boundaries under current consideration. The shading color represents SLA on 16 November 2019. The red dots and red lines represent the wave and wind tracks on 8 October 2019. The black dots and black lines represent the wave and wind tracks on 16 November 2019.

## 5. Summary

A series of sensitivity experiments were carried out using a third-generation numerical wave model to investigate the effect of wind speed and eddy surface currents on CEW.

The results showed that mesoscale eddies can affect wave heights, period, and steepness by 20–30%. Wave direction variability, by contrast, can range from 30°–40°. CEW was more noticeable with strong currents and weak winds, and was governed by wave age ($c_p/W$) and the ratio of wave group velocity to current velocity ($c_g/U$). Wave spectra clearly indicated that the wave energy distribution caused by current mostly happened in the spatial scale of 5–90 km, belonging to the sub-mesoscale and mesoscale. Using CFOSAT-SWIM observations, SWAN-suggested theoretical results were verified. That is, both wave height and steepness were significantly higher at A6 and A7 where WCA > 90° than A2 and A3 where WCA < 90°, which is consistent with the numerical model results. Finally, from the numerical model result and the CFOSAT-SWIM observation wave data, it could be found that along the direction of wave propagation, both wave height and the wave steepness increased on the right side of the anticyclonic eddy in the Northern Hemisphere, while they decreased on the left side (Figure 13). It was identified that wave distribution differences on both sides of the eddy wave data track was perpendicular to the wave propagation direction. The method shown in this work provides a reference for investigating the effect of eddies on wave energy redistribution on a global scale.

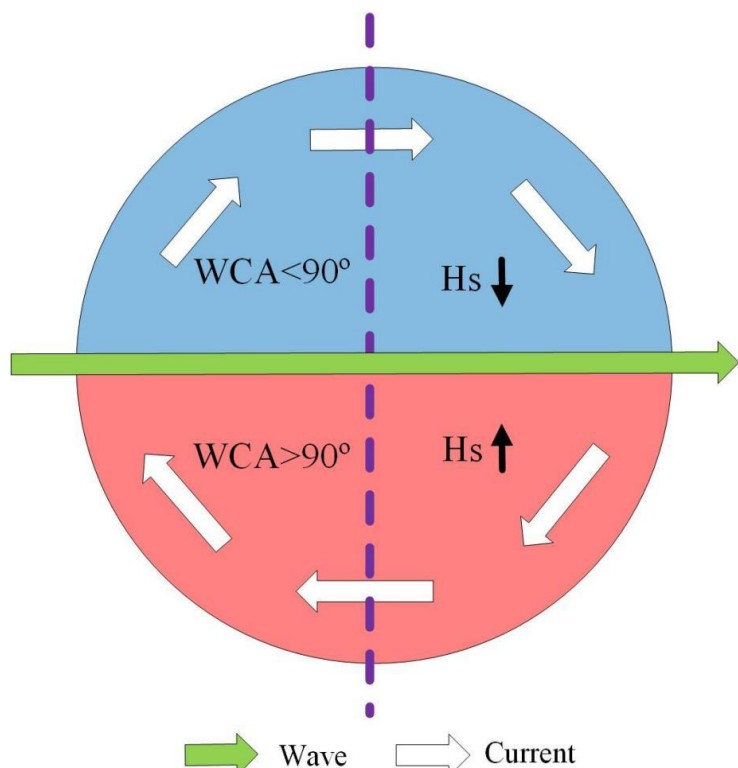

**Figure 13.** The schematic diagram of anticyclonic eddy influence sea surface waves. Blue represents decrease and red represents increase. The green arrow represents wave direction. The white arrow represents current direction. The purple track represents the best wave track selected to study the CEW.

**Author Contributions:** Conceptualization, C.D. and J.W.; methodology, J.W.; software, J.W.; formal analysis, J.W.; investigation, J.W. and B.J.B.; resources, Y.C.; data curation, C.L.; writing—original draft preparation, J.W.; writing—review and editing, J.W. and B.J.B.; visualization, J.W. All authors have read and agreed to the published version of the manuscript.

**Funding:** This research was funded by the National Key Research and Development Program of China, grant numbers 2017YFA0604100 and 2018YFA0605904; and the Natural Science Foundation of Jiangsu Province, grant number BK20180803.

**Institutional Review Board Statement:** Not applicable.

**Informed Consent Statement:** Not applicable.

**Data Availability Statement:** The CFOSAT-SWIM data are collected from the AVISO+ website in France (https://www.aviso.altimetry.fr/home.html, accessed on 1 December 2021) and National Satellite Ocean Application Service in China (https://osdds.nsoas.org.cn, accessed on 1 December 2021). AVISO sea level anomaly Data can be acquired at http://www.aviso.oceanobs.com/ (accessed on 2 November 2021). AVISO geostrophic current anomaly data can be downloaded from https://cds.climate.copernicus.eu/cdsapp#!/dataset/satellite-sea-level-global?tab=overview (accessed on 2 November 2021) and http://www.doi.org/10.11922/sciencedb.01190 (accessed on 2 November 2021). ERA5 reanalysis data can be acquired at https://cds.climate.copernicus.eu/ (accessed on 1 October 2021). The Global Ocean Mesoscale Eddy Atmospheric-Oceanic-Biological Interaction Observational Dataset (GOMEAD)(V1) can be acquired at http://www.doi.org/10.11922/sciencedb.01190 (accessed on 2 November 2021). A detailed description concerning SWAN can be found in the official manual available at http://swanmodel.sourceforge.net/ (accessed on 1 June 2021).

**Acknowledgments:** The authors would like to give our sincerest thanks to the four anonymous reviewers for their constructive suggestions and comments, which are of great value for improving the manuscript. The authors also thank the Editor for the kind assistances and beneficial comments. The authors are grateful for the kind support from the editorial office. We gratefully acknowledge the funders of this study.

**Conflicts of Interest:** The authors declare no conflict of interest.

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
