# Peer review of "Numerical Simulation and Observational Data Analysis of Mesoscale Eddy Effects on Surface Waves in the South China Sea"

_remotesensing, doi:10.3390/rs14061463_

Round 1

Reviewer 1 Report

This paper includes two parts, CFOSAT-SWIM observations of mesoscale eddies and a detailed description of SWAN calculations modeling these observations. The paper's idea, structure, and presentations are well sounded. These results are of current interest from the remote sensing community. I recommend publishing the paper after minor revision and correction of some misprints. Below are comments to improve the paper (see attached file).

Author Response

Thank you for your invaluable comments and suggestions on our manuscript. We have modified the manuscript accordingly, and detailed corrections are listed below point by point. Thanks for your time spent on our manuscript.

Reviewer 2 Report

  1. The download location of some materials is mentioned in the second section, and it is recommended to move directly to the location of "Data Available Statement". In addition, some databases should have versions and DOI codes, which must be added to the manuscript.
  2. Figure 1: The image is deformed and distorted, please re-add the image. Is the current vector in Figure 1 the geostrophic current? I did not see the geostrophic current data in Section 2.1.2, and the current geostrophic current data should be integrated into the CMEMS database. Please write a detailed message about geostrophic current data in section 2.1.2.
  3. Table 1: A horizontal line should be drawn between groups of data, such as Case number 4 and 5.
  4. Figure 2: Why do the spatial scales of the horizontal and vertical axes not indicate [-20 20], and is there any special meaning for the spatial distance of [30-70]? The same problem appears in Figure 3, 4, 5, and 6, and the author should explain it in the manuscript.
  5. Since this study focuses on the case study of the South China Sea, it is strongly suggested that the South China Sea must be added to the title.

Author Response

(The authors gave the same response as above.)

Round 2

Reviewer 2 Report

Dear authors,

Now there is only one small problem, about reply to response 2.

Generally, the data using geostrophic current comes from CMEMS rather than AVISO. It is also possible for the authors to use AVISO, but the detailed version and DOI code must be given so that other researchers can have the opportunity to obtain public data. In addition, the link provided by the authors will lose access on March 22 (https://confluence.ecmwf.int/display/CUSF/CDS+unavailable+on+23+March+2022%3A+access+to+ALL+datasets+and+applications+affected). The author needs to find a way to solve this problem. Perhaps the AVISO data used by the authors can be obtained from CMEMS?

If the author finds a solution and corrects it, the manuscript can be accepted for publication.

Author Response

The AVISO geostrophic current anomaly data (vDT2018) can be obtained from CMEMS at https://cds.climate.copernicus.eu/cdsapp#!/dataset/satellite-sea-level-global?tab=overview, but and while it wont be available on 23 March, 2022, the data can also be acquired from the Global Ocean Mesoscale Eddy Atmospheric-Oceanic-Biological Interaction Observational Dataset (GOMEAD)(V1) at http://www.doi.org/10.11922/sciencedb.01190. This should resolve the issue you have highlighted and we thank the reviewer for the suggestion. We have amended Section 2.1.2 in the revised manuscript.

This manuscript is a resubmission of an earlier submission. The following is a list of the peer review reports and author responses from that submission.

Round 1

Reviewer 1 Report

Please, see the attached document with comments.

Author Response

Thank you for your useful comments and suggestions on the structure of our manuscript. We have modified the manuscript accordingly, and detailed corrections are listed below point by point. Thanks for your time spent on our manuscript.

Reviewer 2 Report

The paper focuses on the effect of annular structure eddies on surface waves. Reported observations of these effects are of importance in Remote Sensing and are subject to publication. But the computational (largest) part of the paper is underdeveloped. A major revision is needed for further consideration of publication.

What is the aim of the computational part of the paper? (Now this part looks like an unstructured set of different issues partially have been presented in the cited articles and repeated with authors calculations) What is the link of this part to the observational (new) results of the paper? These issues must be addressed, and the structure of the computational part must be clarified (maybe with some reduction).

As a pattern of an annular eddy in the waves field in adiabatic approximation (without right part of Equation (1) ) is well known from wave mechanics (see, e.g., Kudryavtsev et al., 1995, and references therein, Mapp et al., 1985, Holthuijsen et al., 1991), the main question is how the wind forcing, nonlinear interactions and wave breaking destroy this pattern resulting in eddy manifestation blurring. The simple modeling approach of this paper could be feasible to consider this question. Then, presenting the results in the framework of the possibility to watch the eddy image, a clear relevance of this part of the paper to remote sensing application could be demonstrated.

Effects of wave development are important for considered scales, but this issue did not address in the paper. The wave age and its effect in model computations did not mention.

What is new in section 4? What new conclusions can be drowning from Figure 10 compared with analogous Figure 6 of Marechal and de Marez, 2021? As this section is very poor and non-constructive, I recommend removing it. I recommend completing the computational part using Figure 10, stating the main features of remote sensing images of eddies before those will be revealed in CFOSAT observation in section 5.

Specific points:

- Subsection 2.1.3. ERA5 wave data: This paragraph does not include a description of wave data. Why it is added to this paper?

- The title of section 3 needs clarifying.

- Equation 2: Why three-wave interactions, bottom friction, and depth-induced wave breaking are accounted for in this paper?

- Table 1: What are initial (or boundary) conditions for computational runs? What are the ages of the waves?

- Wave parameters used need definition (“Surface currents modulate wave fields in terms of Hs, mean wave period (T), average wave steepness (δ)”).

- Equations (4) have errors. The velocity field (4) is not solenoidal and does not correspond to Figure 2.

- Most plates in Figures 3 and 4 are not informative. Change of colormap may improve the figures.

- Subsection 3.3.2: Conservation of absolute frequency and wave action takes place for adiabatic cases.  They do not conserve if the right part of Equation (1) works. This point needs clarifying.

Also, cos(theta) is missed in the equation for omega, where theta is the angle between wavenumber and current velocity.

- Subsection 3.3.3: Wave convergence and wave divergence – what are these? Non-standard terms need definition and explanation of their physical sense.

- The same remark applies to Subsection 3.3.4 and the paragraph after Figure 7.

- The first 6 lines of Section 4 are exact copies of lines 220-225 of the paper by Marechal and de Marez. It is a violation of ethics.

- Subsection 5.1. “Due to the 180° ambiguity of the SWIM wave direction, ERA5 wave data are used to validated and corrected to get the reasonable wave direction.”: This sentence needs clarifying.

Also, the role of ERA5 wave data in this paper must be explained.  

- Equations (7) and (8): What is a physical sense of parameter K or S? How are equations (7) and (8) linked with the fundamental equation (1)?

************

Kudryavtsev, V. N., Grodsky, S. A., Dulov, V. A. and Bol’shakov, A. N. (1995) Observations of wind waves in the Gulf Stream frontal zone, J. Geophys. Res., C10, 100, 20,715-20,727.

Mapp, G. R., C. S. Welch, and J. C. Munday (1985), Wave refraction by warm core rings, J. Geophys. Res., 90(C4), 7153–7162.

Holthuijsen, L. H., and H. L. Tolman (1991), Effects of the Gulf Stream on ocean waves, J. Geophys. Res., 96(C7), 12,755–12,771.

Author Response

Thank you for your useful comments and suggestions on the structure of our manuscript. The link between the numerical simulation and CFOSAT-SWIM observation has been supplemented in the revision. We have modified the manuscript accordingly, and detailed corrections are listed below point by point. Thanks for your time spent on our manuscript.

Round 2

Reviewer 2 Report

This paper includes two well-presented parts, CFOSAT-SWIM observations of mesoscale eddies and a detailed description of SWAN calculations modeling these observations.  Apart from this, the authors discuss the physical phenomenon of the influence of current velocity gradients on wind waves. These considerations are mainly erroneous, include non-traditional terminology, errors in equations, and unsupported claims. Therefore this version of the paper cannot consider a scientific article. I recommend major revision of the paper, hoping that the authors will delete inappropriate parts. Below are comments to improve the paper. (see attached file)

I am not an expert in English, but my mind is that the English of the paper needs a thorough inspection.
